# Cirrhotic Cardiomyopathy Following Bile Duct Ligation in Rats—A Matter of Time?

**DOI:** 10.3390/ijms24098147

**Published:** 2023-05-02

**Authors:** Moritz Uhlig, Marc Hein, Moriz A. Habigt, René H. Tolba, Till Braunschweig, Marius J. Helmedag, Melissa Arici, Alexander Theißen, Axel Klinkenberg, Uwe Klinge, Mare Mechelinck

**Affiliations:** 1Department of Anesthesiology, Faculty of Medicine, RWTH Aachen University, 52074 Aachen, Germany; mhein@ukaachen.de (M.H.); mhabigt@ukaachen.de (M.A.H.); atheissen@ukaachen.de (A.T.); mmechelinck@ukaachen.de (M.M.); 2Institute for Laboratory Animal Science and Experimental Surgery, Faculty of Medicine, RWTH Aachen University, 52074 Aachen, Germany; rtolba@ukaachen.de; 3Department of Pathology, Faculty of Medicine, RWTH Aachen University, 52074 Aachen, Germany; tbraunschweig@ukaachen.de; 4Department of General, Visceral and Transplantation Surgery, Faculty of Medicine, RWTH Aachen University, 52074 Aachen, Germany; mhelmedag@ukaachen.de (M.J.H.); uklinge@ukaachen.de (U.K.); 5Luisenhospital, 52064 Aachen, Germany; melissa.arici@rwth-aachen.de (M.A.); axel.klinkenberg@rwth-aachen.de (A.K.)

**Keywords:** liver cirrhosis, chronic liver failure, rodent model, bile duct ligation, cirrhotic cardiomyopathy, myocardial failure, second hit, mitochondropathy, fibrosis

## Abstract

Cirrhotic patients often suffer from cirrhotic cardiomyopathy (CCM). Previous animal models of CCM were inconsistent concerning the time and mechanism of injury; thus, the temporal dynamics and cardiac vulnerability were studied in more detail. Rats underwent bile duct ligation (BDL) and a second surgery 28 days later. Cardiac function was assessed by conductance catheter and echocardiography. Histology, gene expression, and serum parameters were analyzed. A chronotropic incompetence (P_d31_ < 0.001) and impaired contractility at rest and a reduced contractile reserve (P_d31_ = 0.03, P_dob-d31_ < 0.001) were seen 31 days after BDL with increased creatine (P_d35_, P_d42_, and P_d56_ < 0.05) and transaminases (P_d31_ < 0.001). A total of 56 days after BDL, myocardial fibrosis was seen (P_d56_ < 0.001) accompanied by macrophage infiltration (CD68: P_group_ < 0.001) and systemic inflammation (TNFα: P_group_ < 0.001, white blood cell count: P_group_ < 0.001). Myocardial expression of peroxisome proliferator-activated receptor gamma coactivator 1-alpha (PGC1α) was increased after 31 (P_d31_ < 0.001) and decreased after 42 (P_d42_ < 0.001) and 56 days (P_d56_ < 0.001). Caspase-3 expression was increased 31 and 56 days after BDL (P_d31_ = 0.005; P_d56_ = 0.005). Structural changes in the myocardium were seen after 8 weeks. After the second surgery (second hit), transient myocardial insufficiency with secondary organ dysfunction was seen, characterized by reduced contractility and contractile reserve.

## 1. Introduction

In 2017, 1.5 billion people worldwide were affected by chronic liver failure [1]. Although the causes have changed due to increasing prosperity, particularly in Western developed countries, the mortality rate remains high [2]. Liver cirrhosis represents an advanced stage of progressive liver fibrosis, usually occurring during severe or chronic liver disease. The median survival time of patients with compensated cirrhosis, i.e., without complications, is more than 12 years. However, as soon as the disease state transitions from compensated to decompensated, the average survival of patients is reduced to roughly 2 years [3]. Cardiovascular disease is one of the leading causes of morbidity and mortality in liver cirrhotic patients [4]. Nearly half of all patients with advanced cirrhosis show signs of progressive cardiac dysfunction, characterized by a hyperdynamic circulatory state, a blunted cardiac response to stress, and diastolic dysfunction [5,6,7]. This is summarized under the term cirrhotic cardiomyopathy (CCM) [8,9]. The observed effects on mortality depend on the severity of the systolic and diastolic dysfunction. The implementation of the 2020 diagnostic criteria for CCM led to a reassessment of available data [10]. Overall, CCM was associated with major adverse cardiac events [11]; in particular, elevated filling pressure [11] and reduced global longitudinal strain could be identified as risk factors [12].

Despite a large number of clinical studies, additional but rarer experimental studies could not identify an exact pathomechanism [9]. Certain features of CCM seem to be distinctly different from other forms of cardiomyopathy. For example, bile acids play an important role in the dysregulation of myocardial metabolism (so-called cholecardia), and cannabinoid receptor 1 (CB_1_-R) antagonists were able to immediately restore cardiac function in a rat model [13]. Most studies have investigated functional changes, whereas less is known about structural alterations. Compared to other hypertrophic cardiomyopathies, the presence of late gadolineum enhancement in magnetic resonance imaging as an indicator of irreversible fibrosis is rare. Comparatively higher extracellular volume fractions, which represent reversible fibrosis, decrease significantly after liver transplantation [14].

Although CCM was first reported in 2005 and has been much researched since then, the exact pathomechanism of this multifactorial and exceptional form of cardiomyopathy remains unclear [9,13]. Long-term experimental studies analysing structural and functional changes over time are missing. The results from existing studies are inconsistent. Some describe a significant impairment [15,16] or temporal increase in systolic function at rest [17,18], as well as an increase [18] or reduction in cardiac mass [19,20].

Therefore, we designed an experimental longitudinal study in rats of reasonable length to investigate the development of cardiac pathology. Measurements at different time points should cover both aspects: functional impairment, as well as potential cardiac remodeling. Within four to eight weeks after induction of liver injury by bile duct ligation, cardiac function was described by pressure–volume loops and structure by histology and gene expression.

## 2. Results

The aim of this study was to better understand the effect of liver cirrhosis on cardiac function and structure. For this purpose, 31, 35, 42, and 56 days after bile duct ligation, cardiac function was determined in rats by echocardiography and conductance measurements at rest and under dobutamine stress, followed by histological and molecular biological analysis of the structural cardiac changes.

### 2.1. Animal Model

A total of 89 rats entered the experiment. In total, 6 animals died or were euthanized due to myocardial injury early after the first surgery, 1 animal was euthanized due to bile duct ligation leakage after the first surgery, and 3 animals died due to technical difficulties during the second surgery. Accordingly, 79 animals underwent the entire study protocol and were included in the analysis of this study. The exact number of animals remaining per group and endpoint can be found in Table 1 and Appendix A.

### 2.2. Signs of Liver Injury and Cirrhosis

The analysis of serum samples after BDL showed a significant increase in transaminases (alanine aminotransferase (ALT), aspartate aminotransferase (AST), gamma-glutamyl- transferase (GGT), and alkaline phosphatase (AP) (cf. Figure 1) at all time points examined except for ALT, which showed no significant difference at day 35 and 56. In addition, a continuously increasing bilirubin concentration was observed in the BDL animals, compared to a constant low bilirubin in the control group (cf. Figure 1). There was an additional increase in transaminases on day 31 (ALT, AST, GGT: P_d31_ < 0.01; cf. Figure 1), 3 days after the second intervention, returning to preoperative levels by day 35 after BDL.

### 2.3. Impairment of Myocardial Function

At rest, no significant differences between the groups can be observed for stroke volume (SV), end-systolic volume (VES), end-diastolic volume (VED), end-systolic pressure (PES), or arterial elastance (Ea) (Figure 2 and Figure 3). Heart rate (HR) was decreased in cirrhotic animals only on day 28 and 35 after BDL (P_d28_ < 0.001, P_d35_ = 0.013) and cardiac output (CO) on day 28 (P_d28_ = 0.03). Contractility at rest, displayed by the slope (Mw) of the preload recruitable stroke work, was reduced by 44% at day 31 after induction of cirrhosis compared to control animals (P_d31_ = 0.03; Figure 2).

Dobutamine led to a significant increase in HR in both groups and at all time points except day 31 (P_dob_ < 0.001; Figure 2). The increase in HR in response to dobutamine in the BDL group was significantly lower on day 31 and 56 after BDL (P_d31_ < 0.001, P_d56_ = 0.003), indicating a chronotropic incompetence. Furthermore, dobutamine led to a significant and comparable increase in Mw in both groups from day 35 to 56. In addition, at day 31, the increase in contractility was significantly lower (37 %) in the cirrhotic animals compared to the control group (P_d31_ < 0.001, Figure 2). Dobutamine led to ventricular unloading and a reduced afterload, with significantly lower values of VED, VES, SV, and Ea (P_dob_ < 0.001), while no differences were seen between BDL and controls (cf. Figure 2).

### 2.4. Cardiac Structure

Planimetry of sirius red (SR) staining revealed a progressive increase in collagen fiber content in the myocardium of the cirrhotic animals up to 3.9 ± 0.47% (P_group_ < 0.001, P_time_ < 0.001, P_group*time_ = 0.008). This resulted in the hearts having a significantly higher degree of fibrosis at day 56 after BDL compared to controls (P_d56_ < 0.001, Figure 4). The normalized heart weight and cardiac remodeling index (CRI) showed a non-significant trend on day 35 after BDL and were significantly lower in the cirrhotic animals 42 days after BDL (P_42d_ = 0.026). By day 56 after BDL, the control and BDL became equal again (cf. Appendix A).

The number of CD68-positive cells (cluster of differentiation 68) was significantly higher in the myocardium of the cirrhotic group at all times of examination (cf. Figure 5).

### 2.5. Cardiac Injury and Inflammation

The white blood cell count (WBC) was significantly higher at all time points in the BDL group. While tumor necrosis factor α (TNFα) increased continuously in the BDL group, it decreased in the control group with significant differences at all time points. Furthermore, creatinine was significantly elevated in the BDL group at all time points studied except for day 31 (cf. Table 1). The measured serum markers of myocardial injury from day 28 to 42 showed no increase in BDL animals in comparison to controls (cf. creatinine kinase (CK), creatinine kinase MB-isoenzyme (CKMB), Table 1).

### 2.6. Myocardial Gene Expression

The expression of the analyzed markers for cardiomyopathy: sarcoplasmic/endoplasmic reticulum calcium ATPase 2 (SERCA2), periostin, actin alpha 1 (ACTA1), and connective tissue growth factor (CTGF), natriuretic peptide B (Nppb), the myosin heavy chain α isoform (αMHC), and the myosin heavy chain β isoform (βMHC) showed no relevant differences until day 56 after BDL (cf. Figure 6).

The quantitative polymerase chain reaction (qPCR) for peroxisome proliferator-activated receptor gamma coactivator 1-alpha (PGC1α) showed higher ΔCT values, and thus a reduced gene expression in the myocardium of cirrhotic rats 31 days after BDL (P_d31_ < 0.001, Figure 7a). In contrast, lower ΔCT values, i.e., increased PGC1α gene expressions, were observed 42 days (P_d42_ < 0.001, Figure 7a) and 56 days (P_d56_ < 0.001, Figure 7a) after BDL compared to the respective controls.

Furthermore, the ΔCT values for caspase 3 were significantly lower at day 31 and 56 in the BDL group, indicating a higher gene expression (P_d31_ = 0.005; P_d56_ = 0.005, Figure 7c).

The myocardial expression of cannabinoid receptor 1 (CB_1_-R) was unaltered, and the expression of cannabinoid receptor 2 (CB_2_-R) increased 56 days after BDL in the cirrhotic group (P_d56_ = 0.003, Figure 7d,e). No differences in CB_2_-R between groups were observed from day 31 to 42. Nuclear factor erythroid 2-like 2 (NFE2L2) remained unaltered throughout the observational period (cf. Figure 7b).

## 3. Discussion

Previous studies of CCM have been inconsistent in terms of the mechanism of injury of the liver and follow-up period, and they have often only looked at a short period of time. Thus, the temporal dynamics of CCM could not be well assessed thus far. Therefore, we performed a longitudinal study over eight weeks with four end points (cf. Appendix A).

This study demonstrates for the first time the effect of BDL-induced liver injury on cardiac function and structure in rats over a prolonged period of eight weeks.

Two major observations were made: (a) there was a transient functional myocardial impairment three days after the second surgery that correlated with an impaired mitochondrial function, and (b) there was an isolated myocardial fibrosis without other typical signs of cardiomyopathy after eight weeks.

Three days after the second surgery, cirrhotic animals demonstrated a chronotropic incompetence, reduced contractility at rest, and a reduced contractile reserve compared to controls, as well as additional liver and renal injury (cf. Figure 1). This was accompanied by a transient upregulation of caspase 3 and a downregulation of PGC1α. No changes in cardiac dimensions, relevant fibrosis, or upregulation of fetal genes could be measured at this time. These observations show a high but transient vulnerability of the heart after a second hit (second surgery for the animals) with consecutive secondary organ dysfunction. This is an unexpected effect but nonetheless relevant for the interpretation of existing experimental studies and inpatient management.

These observations appear consistent with the results of Gaskari et al.—in whose experimental setup a controlled hemorrhage after BDL can also be considered as a second hit—which was also followed by chronotropic incompetence in association with decreased contractility [16]. A pronounced myocardial infiltration with monocytes, which in turn are a known source of endocannabinoids [15,16], was noted. In the current investigation, the number of macrophages also remained high during the entire observational period (cf. Figure 5). Endocannabinoids promote cardiac dysfunction by CB_1_-receptor agonism and attenuate inflammation and improve cardiac dysfunction by CB_2_-receptor agonism [19]. Thus, the application of a CB_1_-R antagonist [15] or a CB_2_-R agonist [19] seemed to improve the contractility immediately. These studies reported an unchanged myocardial expression of the CB_1_-R and an irrelevant upregulation of the CB_2_-R. Accordingly, we did not detect any regulation of CB receptors at this time point, but the expression of CB_2_-R at day 56 in our model resembles these findings (Figure 7d,e). The fact that Matyas et al. could almost completely resolve the functional impairment by administration of a CB_2_-R agonist does not seem to be consistent with a chronic change such as CCM but rather suggests a possibly transient functional impairment in which inflammation plays a major role.

Whereas Batkai et al. and Gaskari et al. did not report any structural changes and did not analyze any expressional markers of adverse remodeling, Matyas et al. found cardiac hypotrophy in mice two weeks after BDL with a concomitant decrease in SERCA2 expression and a pathological switch in the MHC isoforms [15,16,19]. In our study, we could not observe a significant decrease in CRI within the BDL group over time. Furthermore, we consistently did not see any changes in the cardiomyopathy markers analyzed (cf. Figure 6). Since Matyas et al. performed their measurements as early as 14 days after induction of liver injury, and thus much earlier than we did, and used mice instead of rats, the PCR data may not be sufficiently comparable [19].The lack of upregulation of markers of adverse remodeling in our study seems consistent with the fact that we could not detect significant fibrosis and hypertrophy before week eight. In part, this could be due to an inhibition of cardiac remodeling by upregulation of PGC1a, which may lead to suppression of CTGF [21]. In summary, it seems unlikely that the observed cardiac functional impairment 31 days after BDL is due to cardiac hypotrophy or fibrosis in the context of cardiac remodeling. The onset of cardiac fibrosis after 8 weeks (cf. Figure 4) might indicate the beginning of adverse remodeling. Liver cirrhosis is well known to lead to a reduced intrathoracic blood volume, lower systemic vascular resistance, and higher CO [7], and the consecutive increased volume or pressure loading of the heart are important triggers of adverse remodeling [22]. As we could not yet observe any hemodynamic stress after 8 weeks (CO, VED; cf. Figure 2d,e), we suppose that typical hemodynamic changes due to liver cirrhosis need more time to develop. Adverse remodeling is likely to continue with CCM progression, and changes in cardiomyopathy markers are expected to occur with prolonged follow-up and increased volume or pressure loading of the heart.

Why Matyas et al. observed a SERCA2 downregulation, and we did not remains speculative. Even though it is known that altered calcium handling is to be expected in CCM, SERCA2 shows an overall group effect (P_group_ = 0.02) but no significant difference at the respective time points in the post hoc analysis, which matches the data of Ward et al. [19,23]. Plausible explanations for this could be either that SERCA2 alterations only become visible as CCM progresses or that, in addition to the regulation of SERCA2, the phosphorylation of phosphoplamban plays an important role in calcium homeostasis in CCM [24]. This inhibits calcium influx into the sarcoplasmic reticulum, which leads to a reduced contractility.

Another important component in fibrogenesis is the RAAS system. It is now known that there are two opposing axes: the angiotensin converting enzyme/angiotensin II/angiotensin 1 receptor pathway, which mediates adverse remodeling, and the angiotensin converting enzyme 2/angiotensin (1–7)/mas receptor pathway, which causes the degradation of collagen and an anti-inflammatory response [25,26]. The balance or imbalance of these two pathways seems to be crucial for the overall effect. In our experiment, TNFα increased strongly on day 56 after BDL (cf. Table 1), which may indicate that at this point the balance shifts towards the angiotensin converting enzyme/angiotensin II/angiotensin 1 receptor pathway, and thus the pro-fibrotic aspects become more apparent, which were previously counterbalanced.

As rats have no gallbladder, the obstruction of the CBD leads to a rapid increase in circulating bile acids. Therefore, a direct effect of circulating bile acids on the myocardium must also be considered. Bile acids have a negative chronotropic and potentially cardiotoxic effect on the myocardium [27,28]. Farnesoid X receptor (FXR) and Takeda G-protein-coupled receptor 5 (TGR5) both have key roles in bile acid signaling and have been found to be expressed on cardiomyocytes (FXR and TGR5), as well as in the vasculature (FXR) [29,30]. Activation of FXR may have proapoptotic effects via activation of caspases 9 and 3 and opening of the mitochondrial permeability transition pore (MPTP) [31]. This is also consistent with the increased caspase-3 expression we observed 31 and 56 days after BDL (cf. Figure 7c).

Another possible explanation for the increased vulnerability of cirrhotic rats could be an impaired energy homeostasis. Desai et al. have already reported that elevated bile acid levels lead to a suppression of PGC1α and thus to an impairment of fatty acid metabolism [32]. Accordingly, we saw a decreased expression of PGC1α (cf. Figure 7a), which may induce mitochondrial dysfunction and thus a higher susceptibility to apoptosis [33]. In line with this, we observed an upregulation of caspase 3, an effector caspase of the intrinsic apoptotic pathway, 31 days after BDL (cf. Figure 7c), suggesting the increased activation of that particular pathway [34]. Additionally, increased TNFα release (cf. Table 1) may cause an increased mitochondrial release of oxygen free radicals and thus oxidative stress via damage to the mitochondrial chain at complex III [35,36]. Garnier et al. previously demonstrated that downregulation of PGC1α plays a key role in the context of congestive heart failure after transverse aortic constriction [37]. Consequently, it is feasible that at least the transient loss of cardiac reserve that we observed after the second hit was due to a PGC1α-mediated adenosine triphosphate (ATP) depletion. Since oxygen was provided to the animals during the surgical procedures and oxygenation was monitored by pulse oximetry (cf. Materials and Methods), it is unlikely that the observed effects were confounded by hypoxia or hypoxic hepatitis. Thus, we hypothesize that elevated bile acid levels and oxidative stress in BDL rats, combined with a second hit, result in suppression of PGC1α and thus decreased ATP synthesis. This transient but significant impairment of cardiac function might be relevant for postprocedural care in patients with liver cirrhosis and might partially account for the reported high acute mortality following surgical interventions [38,39,40].

As already mentioned, the observed cardiac dysfunction regresses spontaneously (Figure 2 and Figure 3). However, we observed a persistent systemic inflammatory state with elevated serum TNFα and WBC levels, as well as macrophage infiltration of the myocardium (Table 1 and Figure 5). The latter might have been due to the high levels of circulating TNFα [41]. Activated macrophages are extremely versatile and play an important role in the regulation of inflammation, matrix remodeling, and fibrosis. They can regulate fibrosis by secreting proteases involved in matrix remodeling [42,43,44]. TNFα-mediated macrophage infiltration of the myocardium is known to lead to fibrotic remodeling [41]. In line with this, we found diffuse myocardial fibrosis involving up to 3.9% of the cross-sectional area (cf. Figure 4). This low grade of fibrosis is consistent with other experimental studies and clinical observations as mentioned in the introduction.

All animals were subjected to the same surgical stress; therefore, the effects of surgery on inflammation are negligible in direct comparison. However, in the control group, the common bile duct was only exposed and mobilized, as is usual in BDL models [45,46,47,48,49]. Since no silk thread was inserted in the control group, it cannot be excluded that part of the inflammatory reaction observed in the BDL group was due to the implanted foreign material, especially as braided silk is a suture material known for an increased foreign-body reaction [50]. However, the use of silk threads in rodents is well established, especially for BDL [45,46,47,48,49]. Furthermore, only very small particles of silk filaments that can be phagocytized seem to result in macrophage-associated cytokine release, and surface activation is not observed [51]. Moreover, cholestasis or liver cirrhosis itself is known to lead to a pronounced systemic inflammatory response [52,53]; therefore, it is unlikely that the observed inflammatory response is mainly caused by the silk thread. Nevertheless, this is a limitation of the model.

Desai et al. have previously shown that an upregulation of PGC1α can restore a dysfunctional fatty acid oxidation in cardiomyocytes [32]. It could be conceivable that the myocardial failure seen on day 31 triggered an upregulation of PGC1α via the adenosine monophosphate-activated protein kinase (AMPK) to stabilize myocardial energy balance [54,55]. In line with this, we saw an insignificant trend on day 35 and a significant upregulation of PGC1α at day 42 and 56 after BDL. Hence, we assume that a PGC1α upregulation compensated for a dysfunctional energy homeostasis [32,55].

The reported gene-expression pattern of PGC1α indicates mitochondropathy but is not definitive. Nevertheless, there is already some evidence indicating mitochondropathy in the context of cholecardia [32,56]. Since the transient increased vulnerability for a second hit was an unexpected result, differentiated methods to reveal the mitochondropathy in this study are lacking. More detailed studies are needed to conclusively assess whether CCM can be explained, at least in part, by mitochondrial dysfunction.

Although we had already chosen a relatively long follow-up period, we saw the first structural changes, which might be the first signs of CCM, no earlier than eight weeks but without relevant functional impairment. This would imply that a further extension of the observational period might have been necessary to eventually observe a decompensation of this fragile balance, and thus an actual cirrhotic cardiomyopathy with consecutive functional impairment, and not only an increased susceptibility to a second hit.

Even though Batkai et al. had a similarly long follow-up period of 10–12 weeks in their carbon tetrachloride (CCL_4_)-based model, the cardiac impairment appeared to be more pronounced [15]. This may be due either to the two- to four-weeks-longer duration of the experiment or to the underlying etiologies of CCM which may play a larger role than assumed thus far [9,57,58]. Therefore, a possible explanation could also be that there are different phenotypes of CCM.

In summary, cirrhotic rats showed a pronounced vulnerability to a second hit with a transient but significant impairment of contractility and secondary organ dysfunction. We hypothesize that this is due to a dysfunctional energy homeostasis. Furthermore, the results indicate that CCM in Sprague–Dawley rats develops at the earliest about eight weeks after BDL with structural changes and without functional impairments. Experimental protocols and time are certainly relevant key factors for the observed effects and could explain discrepancies with other studies.

## 4. Materials and Methods

### 4.1. Animal Model/Protocol

This study was designed to answer two different research questions in one approach, in order to reduce the number of animals according to the 3Rs principle of Russell and Burch [59]: the effect of liver cirrhosis on cardiac function and structure and, at the same time, its effect on vascular remodeling after balloon injury of the carotid artery, which was beyond the focus of the current investigation and will therefore be published separately.

All experiments were performed on male Sprague–Dawley rats (RjHan:SD; Janvier Labs, Le Genest-Saint-Isle, France) weighing 489.8 g ± 29.07 g on average. Animals were housed in an environmentally controlled room with a 12-h light/dark cycle with food and water available ad libitum. Experiments were preceded by 7 days of acclimation, and no intervention was performed during this period. All experimental procedures were within the German Animal Welfare Act (§ 8 Abs. 1, Tierschutzgesetz [60]) and were approved by the governmental Animal Care and Use Office (No 84-02.04.2016. A391, Landesamt für Natur, Umwelt und Verbraucherschutz Nordrhein-Westfalen, Recklinghausen, Germany).

The animals were visited at least once daily during the entire study, to assess the rats’ general condition using a semiquantitative score sheet [61]. In the score, body weight, general condition, spontaneous behavior, and behavior-specific criteria were considered (5–9 points = low stress, 10–19 points = moderate stress, and ≥20 points = high stress). In cases where the animals had more than 20 score points or 10–19 points for more than 72 h, animal welfare officers were instantly consulted, and the animals were immediately euthanized, if necessary.

#### 4.1.1. First Surgery

Liver cirrhosis was provoked by ligation of the common bile duct (CBD), in accordance with Tag et al. [62]. Analgesia was achieved by the subcutaneous administration of buprenorphine (0.01–0.03 mg/kg body weight, Temgesic, Essex Pharma GmbH-Msd Sharp and Dohme GmbH, Haar, Germany) 30 min before surgery. Anesthesia was induced with 4 vol% isoflurane until loss of consciousness and maintained with 2 vol% in 100% oxygen with spontaneous breathing. Adequate depth of anesthesia was ensured by testing interdigital reflexes. Next, a first transthoracic echocardiography (TTE) was performed. The abdominal cavity was opened via a median laparotomy followed by dissection of the CBD. It was ligated twice with a silk thread 5/0 (18020-50, Fine Science Tools, Vancouver, Canada) and transected between ligations. Towards the end of the surgery, the wound margin was infiltrated with ropivacaine (Ropivacaine Kabi 10 mg/mL, Fresenius Kabi AG, Kriens, Switzerland) 0.5% (25 mg/kg BW). In addition, subcutaneous application of dipyrone (100 mg/kg BW, diluted to 100 mg/mL; novaminsulfon ratio 1 g/2 ml, Ratiopharm, Ulm, Germany) was performed 30 min before the end of anesthesia, 6 h after surgery, and thereafter as needed.

Sham-operated animals served as controls, with the same procedure without ligation and transection of the CBD. Allocation to the respective intervention groups was performed in a quasi-randomized manner, as the BDL animals were already visually distinguishable a few days after the first surgery due to the onset of jaundice with markedly elevated bilirubin levels. This resulted in a yellowish discoloration of the skin of the Sprague–Dawley rats [45].

#### 4.1.2. Second Surgery

As part of the study on carotid arterial remodeling, balloon dilation of the left carotid artery was performed four weeks after BDL with the same anesthesiologic management, as described above. Prior to surgery, TTE was performed after the induction of general anesthesia, and a blood sample was taken via a central venous catheter.

#### 4.1.3. Final Surgery

A total of 3, 7, 14, or 28 days after balloon dilation, which corresponds to 31, 35, 42, and 56 days after the first surgery, a third and final anesthesia was induced to perform a TTE and registration of pressure–volume (PV) loops with a conductance catheter at rest and during dobutamine stress, blood sampling, and organ (heart, liver) harvesting. Rats were subsequently euthanized under deep anesthesia by exsanguination combined with heart removal. The blood was collected for biochemical analysis, as described later. Heart weight was measured and normalized to tibia length to describe the degree of hypertrophy.

### 4.2. Hemodynamics

In order to assess cardiovascular function, basic monitoring, echocardiographic examination, and conductance catheter measurements were performed at rest and under dobutamine-induced stress on day 28 to 56 using an infusion of 10 μg/kg/min dobutamine. Dobutamine allowed assessment of the contractile reserve, which is typically compromised in CCM before the cardiac function at rest is impaired [10].

#### 4.2.1. Basic Monitoring

Following induction of anesthesia, rats were positioned supine and monitoring was established, consisting of pulse oximetry for blood oxygen saturation (SpO_2_) measurement, electrocardiography (ECG), and noninvasive measurement of blood pressure (NIBP). Pulse oximetry was applied to the paw of the rodents using a commercially available infrared system (Radical-7, Masimo, Puchheim, Germany). A 3-lead Eindhoven/Goldberger ECG was derived using needle electrodes and a differential amplifier (BioAmp FE231, ADInstruments Ltd., Oxford, UK). In addition, NIBP was measured during first and second surgery with a commercially available device (IN125/R, ADInstruments Ltd., Oxford, UK) using an inflatable tail cuff in combination with a pulse sensor, to detect hypotension, as a consequence of hypovolemia or very deep anesthesia. Maintenance of adequate body temperature was ensured by a feedback-controlled heating pad (TCAT-2LV controller, Physitemp, Clifton, NJ, USA) with a rectal probe.

All parameters were measured and recorded throughout every procedure and stored for further analysis using Labchart (ADInstruments Ltd., Oxford, UK).

#### 4.2.2. Echocardiography

Prior to each surgery, a TTE was performed to calibrate the volume signal of the conductance catheter and describe the global dimensions and function on day 0, 28, 31, 35, 42, and 56: a 10 MHz transducer (GE 10S-RS, GE Healthcare, Chicago, IL, USA) connected to a Vivid I (GE Healthcare, Chicago, IL, USA) was used to record a sequence in the parasternal long axis (LAX). Data analysis was performed with EchoPAC software (version 201; GE Healthcare, Chicago, IL, USA). End-diastolic (VED) and end-systolic volumes (VES) were determined following the modified Simpson’s method [19]. Stroke volume (SV) and cardiac output (CO) was calculated from heart rate (HR), VED, and VES. The cardiac remodeling index (CRI) was calculated as a quotient of the echocardiographically measured left ventricular mass and the tibial length [63].

#### 4.2.3. Pressure Volume Loops

For advanced diagnostics of cardiac function, a 2F conductance catheter (SPR-869, Millar Instruments, Houston, TX) was retrogradely advanced via the right external carotid artery (ECA) into the left ventricle to record pressure–volume loops on day 31, 35, 42, and 56 [64]. The correct position of the catheter was verified by TTE and the shape of the obtained PV loop. Derived hemodynamic parameters were calculated after two-point calibration of the volume signal with VED and VES using Lab Chart (Lab Chart Pro 8.1, ADInstruments Ltd., Oxford, UK). From recordings under stable conditions, the end-systolic pressure (PES) and arterial elastance (Ea = PES/SV) were calculated.

In order to determine contractility, the inferior vena cava was accessed by a small subxiphoid median laparotomy and compressed five times with a cotton swab [64]. Changes in stroke work (SW) related to VED were calculated from PV loops obtained during loading intervention: the slope “Mw” of this so-called preload recruitable stroke work reflects a load-independent parameter of contractility [65].

### 4.3. Histological Staining

All tissue samples were fixed in 4% buffered formalin (ROTI Histofix 4%, Carl Roth GmbH + Co. KG, Karlsruhe, Germany) for one week, then embedded in paraffin and cut into 3 µm sections. Prepared slides were then dewaxed and rehydrated using standard xylol and a descending alcohol series.

Sections of liver and the midpapillary part of the left ventricle were stained with hematoxylin and eosin staining (HE; Merck KGaA, Darmstadt, Germany), according to a standard protocol.

Myocardial fibrosis was analyzed using sirius red staining (SR; Direct Red 80 (Cat no. 365548; Sigma-Aldrich Chemie GmbH, Taufkirchen, Germany) on one slide per animal. Collagen content was quantified in relation to the total area. This was determined using an automated and standardized analysis of sirius red staining under polarized light with the bioimage analysis software QuPath (QuPath Version: 0.2.3, QuPath developers, University of Edinburgh) [66].

For immunohistological staining, the ZytoChem-Plus AP Polymer Kit (Zytomed Systems GmbH, Berlin, Germany) was used according to the manufacturer’s protocol. Macrophages in cardiac sections were detected using a cluster of differentiation 68 (CD68) antibody (Cat. No. BM4000S, OriGene Technologies GmbH, Herford, Germany) as the primary antibody. CD68-positive cells were then counted in 10 high power fields (HPF; 40x, 2560 × 1440 pixels) per slide (one slide per animal) in a blinded manner.

### 4.4. Serum Analysis

Whole blood samples were collected from the rats. A small amount was used to determine the white blood cell (WBC) count using a cell counter (Celltac Alpha VET MEK-6550, Nihon Kohden, Tokyo, Japan) immediately after sample collection.

For serum collection, whole blood was pipetted into serum micro sample tubes (Micro sample tubes Serum Gel, 1.1 mL, push cap, EU/ISO, SARSTEDT AG and Co. KG, Nümbrecht, Germany) and centrifuged at a relative centrifugal force of 5000× *g* for 10 min at 4 °C to separate the solid and liquid components. The supernatant (the serum) was then pipetted into aliquots. All blood analyses except for the enzyme-linked immunosorbent assay (ELISA) measurements were performed directly; for the ELISA measurements, the serum was immediately stored at −80 °C for a later joint analysis of all samples. In cases of analysis after storage, the thawed samples were used only once.

In order to ascertain whether BDL induced a significant liver injury, alanine aminotransferase (ALT), aspartate aminotransferase (AST), and total bilirubin were measured. In order to assess inflammatory processes, tumor necrosis factor α (TNFα) was measured with an ELISA kit from R&D Systems (Rat TNF-alpha Quantikine ELISA Kit, RTA00; R&D Systems, Inc. Minneapolis, MN, USA). All assays were performed according to the respective manufacturers’ protocols from serum obtained as described above.

Creatine kinase (CK) and creatine kinase MB-isoenzyme (CKMB) were determined by the Laboratory of Hematology at the Institute of Laboratory Animal Science and Experimental Surgery, RWTH Aachen University, Faculty of Medicine, Aachen, Germany from serum obtained as described above.

### 4.5. Quantitative Polymerase Chain Reaction

The expression of markers for adverse remodeling, apoptosis, and mitochondrial dysfunction was measured in cardiac specimens from basal and apical parts of the left ventricle using quantitative polymerase chain reaction (qPCR). Total ribonucleic acid (RNA) was extracted using a commercially available RNA/protein extraction kit (NucleoSpin^®^ RNA/Protein, Machery-Nagel, Düren, Germany). RNA was reverse transcribed to complementary deoxyribonucleic acid (cDNA) using a high-capacity reverse transcription kit (Thermo Fisher Scientific, Langerwehe, Germany). A PCR reaction was performed using cDNA, a Taqman^®^ Gene Expression Master Mix (Thermo Fisher Scientific, Langerwehe, Germany), and specific TaqMan^®^ probes.

As markers of adverse cardiac remodeling probes for natriuretic peptide B (NPPB, Rn00580641_m1), the myosin heavy chain α isoform (Myh6, Rn00568304_m1), and the myosin heavy chain β isoform (Myh7, Rn00568328_m1) were analyzed [67,68]. In order to assess adverse remodeling, periostin (Postn, Rn01494627_m1) actin alpha 1 (Acta1, Rn01641150_m1), and connective tissue growth factor (Ctgf, Rn00573960_g1) were measured [69]; in order to assess calcium handling, sarcoplasmic/endoplasmic reticulum calcium ATPase 2 (Serca2, Rn00568762_m1) was used [70]. Expression of caspase 3 (Casp3, Rn00563902_m1) was analyzed to assess apoptosis. As previous studies have suggested that cannabinoid receptor 1 (Cnr1, Rn02758689_s1) and cannabinoid receptor 2 (Cnr2, Rn04342831_s1) may play an important role in CCM, they were quantified with the according probes [19,71]. Mitochondrial dysfunction was displayed by expression of the key regulator peroxisome proliferator-activated receptor gamma (Ppargc1a, Rn00580241_m1) [72] and nuclear factor erythroid 2-related factor 2 (Nfe2I2, Rn00477784_m1) to assess the antioxidative response [73]. Glyceraldehyde-3-phosphate dehydrogenase (Gapdh, Rn99999916_s1) on a StepOne-Plus^®^ Cycler (Thermo Fisher Scientific, Langerwehe, Germany) was used as the house-keeping gene. The gene expression levels were displayed as differences of CT values between the gene of interest and the reference gene (ΔCT).

### 4.6. Statistics

Statistical analysis and presentation of data were performed using GraphPad Prism 9 (version 9.2.0, GraphPad Software, Inc., La Jolla, CA) and JMP Pro 15.2.1 (SAS Institute, Cary, NC) and SAS software 9.4 (SAS Institute, Cary, NC, USA).

Data are presented as the mean ± standard error of the mean (SEM) or as Tukey’s boxplots. The normality and homoscedasticity of data were checked by diagnostic plots (residuals and quantile plots). Data were transformed where necessary either using logarithmic or Box–Cox transformation. Overall testing was performed using a generalized linear mixed model analysis considering group and time for serum parameters and group, time, and the dobutamine effect for hemodynamic measurements. In the case of heteroscedasticity, the heterogenous variances were accounted for in the model. All *p*-values from parametric analysis were adjusted for multiple comparisons using the Shaffer-simulated approach.

Where the distribution of the data was unknown, a nonparametric test was chosen (e.g., Kruskall–Wallis) and the resulting *p*-values were adjusted for multiple comparisons using FDR (false discovery rate) [74]. The null hypothesis was rejected when *p* < 0.05.

## Figures and Tables

**Figure 1 ijms-24-08147-f001:**
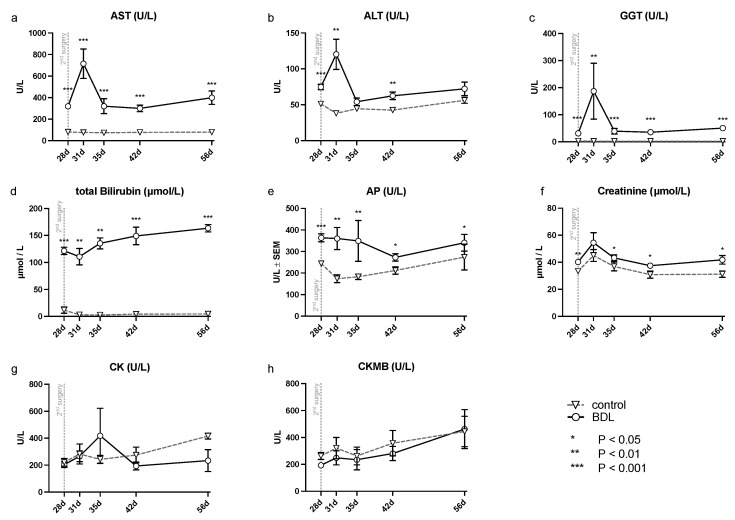
Measured serum parameters for hepatic, renal and cardiac injury in rats after bile duct ligation (BDL) versus control over time. As parameters of hepatic injury, (**a**) aspartate aminotransferase (AST), (**b**) alanine aminotransferase (ALT), (**c**) gamma-glutamyl- transferase (GGT), (**d**) total bilirubin and (**e**) alkaline phosphatase (AP) were assessed. In order to detect renal injury, (**f**) creatinine was analyzed, and for cardiac injury, (**g**) creatinine kinase (CK) and the (**h**) creatinine kinase MB-isoenzyme (CKMB) were analyzed using a Kruskal–Wallis test corrected for multiple comparisons using the false discovery rate (FDR). (* *p* < 0.05, ** *p* < 0.01, *** *p* < 0.001; Mean ± standard error of the mean (SEM)). *p* ≤ 0.05 was considered significant.

**Figure 2 ijms-24-08147-f002:**
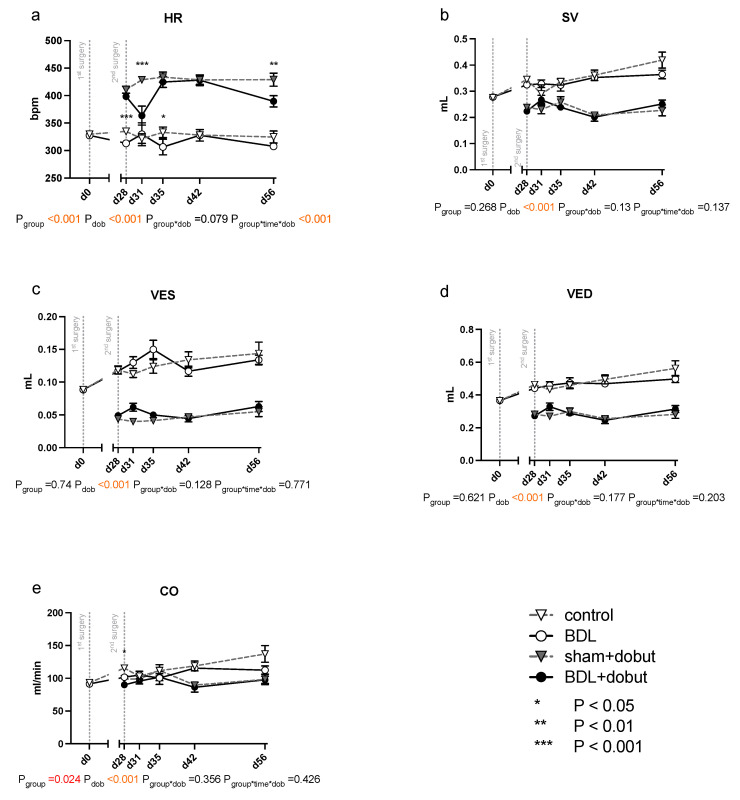
Echocardiographic measurements of cardiac function in rats after bile duct ligation (BDL) versus control over time. As parameters of cardiac function, (**a**) heart rate (HR), (**b**) stroke volume (SV), (**c**) end-systolic (VES) and (**d**) end-diastolic (VED) volumes, and (**e**) cardiac output (CO) were measured at rest and under dobutamine stress for controls and bile-duct-ligated (BDL) rats. Data were analyzed using a generalized mixed model analysis corrected for multiple comparisons using the Shaffer-simulated approach (* *p* < 0.05, ** *p* < 0.01, ****p* < 0.001; Mean ± standard error of the mean (SEM)). The interaction effect of group and dobutamine is expressed as group*dob and the interaction of group, time and dobutamine as group*time*dob. *p* ≤ 0.05 was considered significant.

**Figure 3 ijms-24-08147-f003:**
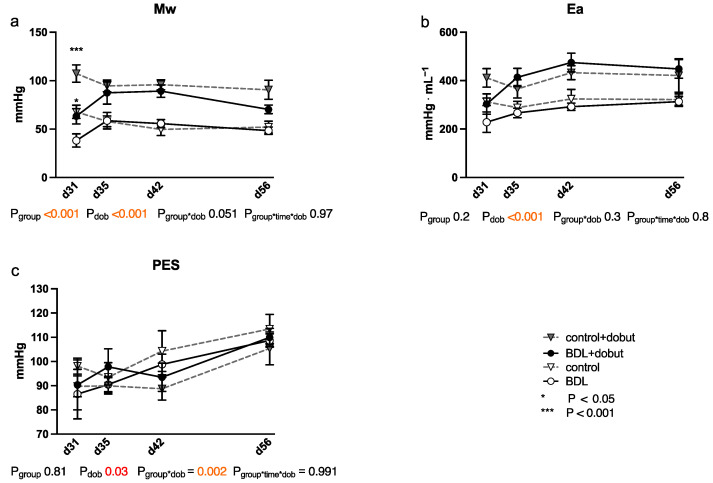
Cardiac function measured using a conductance catheter in rats after bile duct ligation (BDL) compared with controls over time. The measured parameters were (**a**) the slope (Mw) of the preload recruitable stroke work, (**b**) arterial elastance (Ea), and (**c**) end-systolic pressure (PES). Data were analyzed using a generalized mixed model analysis corrected for multiple comparisons using the Shaffer-simulated approach (* *p* < 0.05, *** *p* < 0.001; Mean ± standard error of the mean (SEM)). The interaction effect of group and dobutamine is expressed as group*dob and the interaction of group, time and dobutamine as group*time*dob. *p* ≤ 0.05 was considered significant.

**Figure 4 ijms-24-08147-f004:**
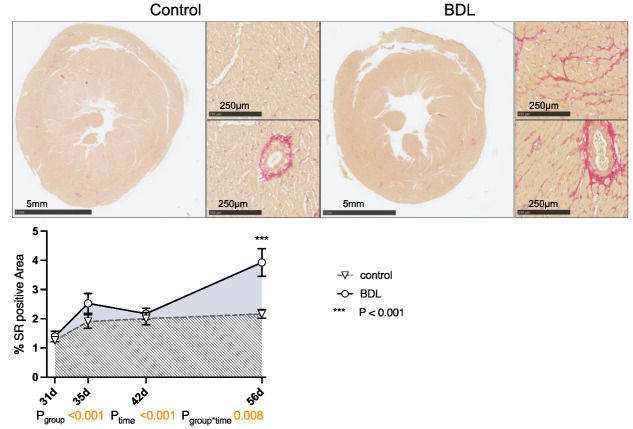
Sirius red (SR) staining of rat myocardium after bile duct ligation (BDL) versus control. Exemplary slices from control and BDL animals after 56 days are displayed as overview and 10× magnification of left ventricular myocardium respectively. The percentage area of positive SR staining of the entire cross section of the heart was quantified from day 31 to 56 displayed as mean ± standard error of the mean (SEM) (P_group_ < 0.001, P_time_ < 0.001, P_group*time_ = 0.008, *** P_d56_ < 0.001). Data were analyzed using a generalized mixed model analysis corrected for multiple comparisons using the Shaffer-simulated approach. The interaction effect of group and time is expressed as group*time. *p* ≤ 0.05 was considered significant.

**Figure 5 ijms-24-08147-f005:**
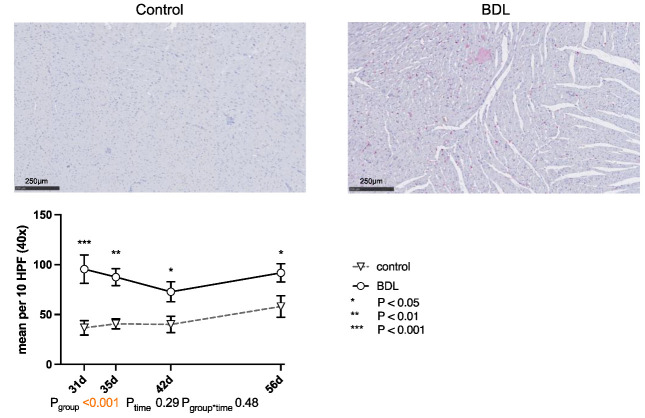
Immunohistochemical staining of CD68 positive cells in rat myocardium after bile duct ligation (BDL) versus controls. Exemplary histologic images are shown for both groups at day 31 after BDL. Scale bars correspond to 250 µm length. Positive cells were counted in 10 high power fields (HPF, 40×; 2560 × 1440 pixels) and displayed as mean ± standard error of the mean (SEM). (*** P_d31_ < 0.001; ** P_d35_ = 0.002; * P_d42_ = 0.017, P_d56_ = 0.019, P_group_ < 0.001, P_time_ = 0.29, P_group*time_ = 0.48). Data were analyzed using a generalized mixed model analysis corrected for multiple comparisons using the Shaffer-simulated approach. The interaction effect of group and time is expressed as group*time. *p* ≤ 0.05 was considered significant.

**Figure 6 ijms-24-08147-f006:**
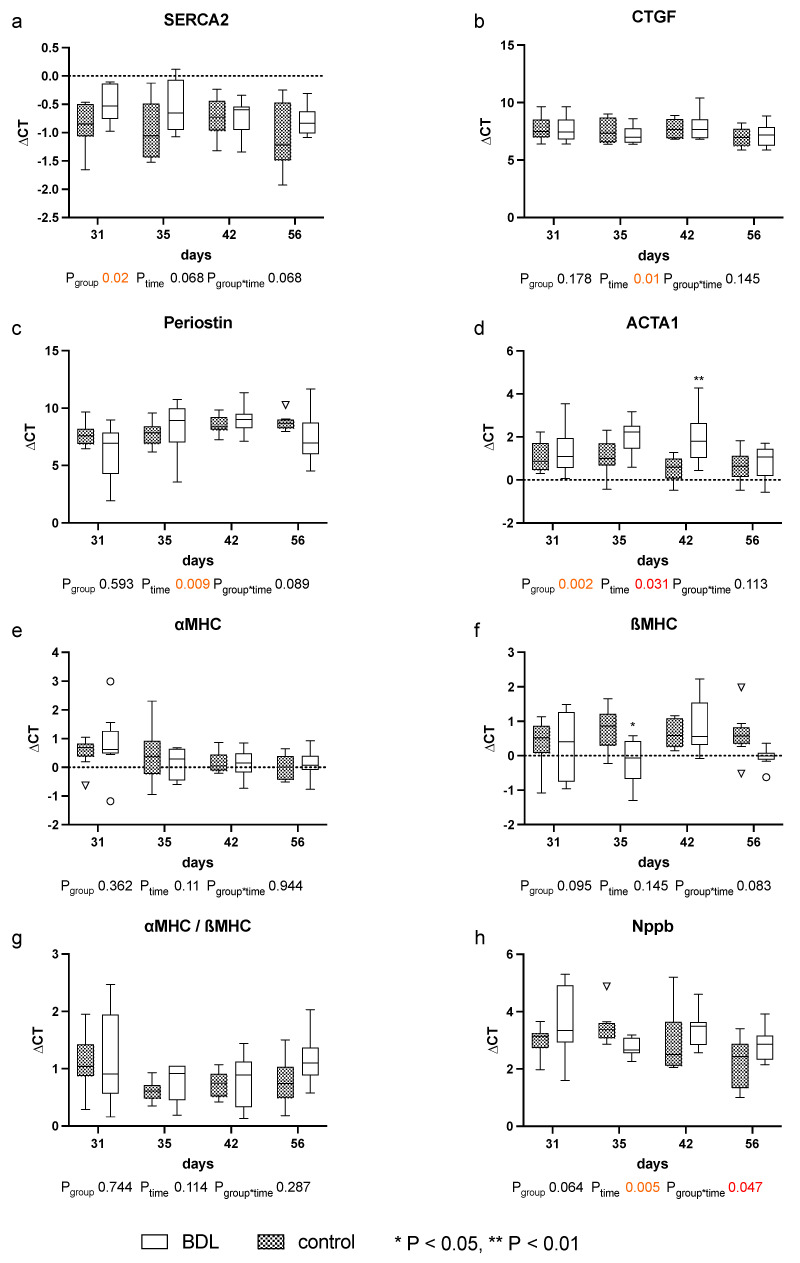
Gene expression of markers for cardiomyopathy in myocardium of rats after bile duct ligation (BDL) versus controls. Displayed are the ΔCT (cycle time) values for (**a**) SERCA2, (**b**) CTGF, (**c**) Perisotin, (**d**) ACTA1, (**e**) αMHC, (**f**) βMHC, (**g**) αMHC/βMHC and (**h**) Nppb. (* *p* < 0.05, ** *p* < 0.01; Data presented as Tukey’s boxplots). Data were analyzed using a generalized mixed model analysis corrected for multiple comparisons using the Shaffer-simulated approach. The interaction effect of group and time is expressed as group*time. *p* ≤ 0.05 was considered significant.

**Figure 7 ijms-24-08147-f007:**
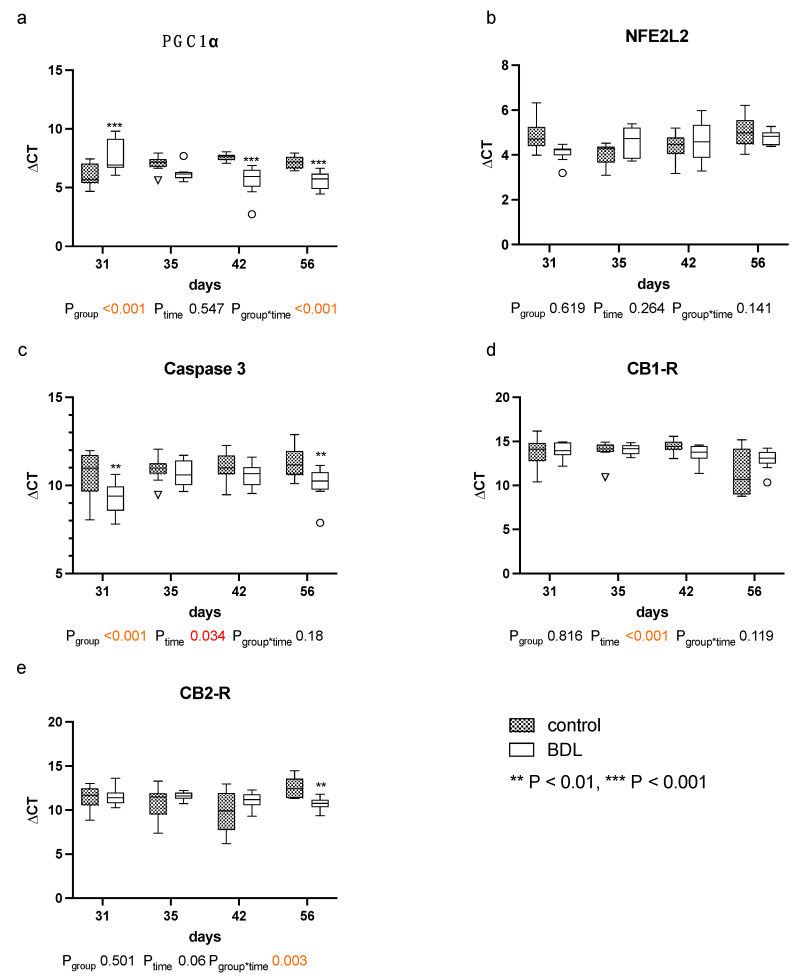
Gene expression of markers for mitochondrial dysfunction, apoptosis and cannabinoid receptors: in myocardium of rats after bile duct ligation (BDL) versus controls. Displayed are the ΔCT (cycle time) values for (**a**) peroxisome proliferator-activated receptor gamma coactivator 1-alpha (PGC1α), (**b**) Nuclear factor erythroid 2-related factor 2 (NFE2L2), (**c**) Caspase 3, (**d**) Cannabinoid Receptor 1 (CB_1_-R) and (**e**) cannabinoid receptor 2 (CB_2_-R). (** *p* < 0.01, ****p* < 0.001; Data presented as Tukey’s boxplots). Data were analyzed using a generalized mixed model analysis corrected for multiple comparisons using the Shaffer-simulated approach. The interaction effect of group and time is expressed as group*time. *p* ≤ 0.05 was considered significant.

**Table 1 ijms-24-08147-t001:** Serum markers of cardiac and renal impairment and inflammation in rats after bile duct ligation (BDL) compared to control over time.

Days after BDL		28 Days	31 Days	35 Days	42 Days	56 Days	P_time_	P_group*time_	P_group_
n Control/BDL		*n* = 40/39	*n* = 10/10	*n* = 10/9	*n* = 10/10	*n* = 10/10			
CK	Control	221 ± 28	283 ± 74	243 ± 30	275 ± 59	416 ± 22	0.055	0.58	0.26
	BDL	204 ± 22	266 ± 40	418 ± 204	194 ± 30	234 ± 81			
	P_group_	0.99	0.99	0.95	0.81	0.60			
CKMB	Control	262 ± 26	321 ± 80	262 ± 66	358 ± 94	445 ± 113	0.053	0.91	0.34
	BDL	193 ± 19	248 ± 53	234 ± 76	280 ± 53	462 ± 145			
	P_group_	0.15	0.93	0.99	0.99	0.99			
Crea	Control	33.3 ± 1.2	45 ± 4.4	36.1 ± 3	31 ± 2.4	34.1 ± 2.5	non-parametric test	non-parametric test	non-parametric test
	BDL	40 ± 1.6	52.9 ± 6.7	43.3 ± 2.2	37.5 ± 1.7	44.4 ± 2.3			
	P_group_	0.006	0.53	0.024	0.013	0.013			
WBC	Control	11 ± 1	10 ± 1	11 ± 1	9 ± 1	9 ± 1	0.79	0.28	<0.001
	BDL	26 ± 2	24 ± 2	28 ± 7	28 ± 4	35 ± 5			
	P_group_	<0.001	<0.001	<0.001	<0.001	<0.001			
TNFa	Control	2.21 ± 0.35	0.89 ± 0.36	1.25 ± 0.38	0.05 ± 0	1.22 ± 0.83	0.01	<0.001	<0.001
	BDL	4.4 ± 0.36	7.22 ± 2.76	5.7 ± 0.97	8.12 ± 2.73	20.1 ± 3.9			
	P_group_	<0.001	<0.001	<0.001	<0.001	<0.001			

Serum markers for cardiac injury were creatinine kinase (CK) and creatinine kinase MB-isoenzyme (CKMB), for inflammation these were white blood cell count (WBC) and tumor necrosis factor α (TNFα), and for renal dysfunction creatinine was used. Data were analyzed using a general linear mixed model analysis corrected for multiple comparisons using the Shaffer-simulated approach. The interaction effect of group and time is expressed as group*time. Creatinine (Crea) data was analyzed using a Kruskal-Wallis test corrected for multiple comparisons using the false discovery rate (FDR). *p* ≤ 0.05 was considered significant and the data were shown as mean ± standard error of the mean (SEM).

## Data Availability

The raw data of the laboratory values generated and analyzed during the current study are available from the corresponding author on reasonable request.

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
