# Peer review of "Cirrhotic Cardiomyopathy Following Bile Duct Ligation in Rats—A Matter of Time?"

_ijms, 2023, doi:10.3390/ijms24098147_

Round 1
Reviewer 1 Report
Cirrhotic patients often suffer from cirrhotic cardiomyopathy (CCM), and previous animal models of CCM were inconsistent concerning time and mechanism of injury. In response to this, temporal dynamics and cardiac vulnerability were studied in more detail in this manuscript, and draw conclusions: CCM corresponds more to a function than to a structural disorder and with symptoms distinctly different from those of other cardiomyopathies. The question posed by the authors is new, and the results are relevant and accurate. Therefore, I think that this manuscript is worth publication.
Author Response
We would like to sincerely thank Reviewer 1 for taking the time to read and review our study. We are very glad to hear that he likes the article at hand and considers it suitable for publication. As suggested, we have spell-checked the paper again and checked the references.
Reviewer 2 Report
The authors have provided a novel manuscript describing a rat model for cirrhotic cardiomyopathy. I don’t have any significant concerns regarding the data in the paper and their conclusions. But some issues need to be addressed before accepting this paper.
General comments:
1) Authors have widely used acronyms in the study, but they define them randomly in the manuscript, which is not the convention. They should describe them in the first instance where it is used.
2) In general, the manuscript could use some more version of editing, the flow of the paper is quite disruptive, and most sections have little to no introduction before they are presented. Why was a particular dobutamine treatment done, or why did they choose genes for qPCR rather than others reported in the literature?
3) The authors should discuss why there was no change in markers of cardiomyopathy in their model.
4) The authors have performed two surgical innervations. They should mark them in the figure, making them more intuitive.
5) They present the data for dobutamine-treated groups in histopathology, hypertrophy, and qPCR results. Was there a change in fibrosis levels in dobutamine-treated animals?
6) Sirius red staining could be presented as a whole mount stain or describe which section of the heart was chosen for imagining and why.
7) qPCR results are presented as delta CT values; fold change would be more intuitive.
8) In line 230, the phrase “structural origin” is ambiguous. Does it mean changes in the ultra-structure of cardiomyocytes or the hypertrophy of the whole organ?
